

# Effect of glucocorticoid blockade on inflammatory responses to acute sleep fragmentation in male mice

Zim Warda Hasan, Van Thuan Nguyen and Noah T. Ashley

Department of Biology, Western Kentucky University, Bowling Green, KY, United States of America

## ABSTRACT

The association between sleep and the immune-endocrine system is well recognized, but the nature of that relationship is not well understood. Sleep fragmentation induces a pro-inflammatory response in peripheral tissues and brain, but it also activates the hypothalamic-pituitary-adrenal (HPA) axis, releasing glucocorticoids (GCs) (cortisol in humans and corticosterone in mice). It is unclear whether this rapid release of glucocorticoids acts to potentiate or dampen the inflammatory response in the short term. The purpose of this study was to determine whether blocking or suppressing glucocorticoid activity will affect the inflammatory response from acute sleep fragmentation (ASF). Male C57BL/6J mice were injected i.p. with either 0.9% NaCl (vehicle 1), metyrapone (a glucocorticoid synthesis inhibitor, dissolved in vehicle 1), 2% ethanol in polyethylene glycol (vehicle 2), or mifepristone (a glucocorticoid receptor antagonist, dissolved in vehicle 2) 10 min before the start of ASF or no sleep fragmentation (NSF). After 24 h, samples were collected from brain (prefrontal cortex, hypothalamus, hippocampus) and periphery (liver, spleen, heart, and epididymal white adipose tissue (EWAT)). Proinflammatory gene expression (TNF-$\alpha$ and IL-1$\beta$) was measured, followed by gene expression analysis. Metyrapone treatment affected pro-inflammatory cytokine gene expression during ASF in some peripheral tissues, but not in the brain. More specifically, metyrapone treatment suppressed IL-1$\beta$ expression in EWAT during ASF, which implies a pro-inflammatory effect of GCs. However, in cardiac tissue, metyrapone treatment increased TNF-$\alpha$ expression in ASF mice, suggesting an anti-inflammatory effect of GCs. Mifepristone treatment yielded more significant results than metyrapone, reducing TNF-$\alpha$ expression in liver (only NSF mice) and cardiac tissue during ASF, indicating a pro-inflammatory role. Conversely, in the spleen of ASF-mice, mifepristone increased pro-inflammatory cytokines (TNF-$\alpha$ and IL-1$\beta$), demonstrating an anti-inflammatory role. Furthermore, irrespective of sleep fragmentation, mifepristone increased pro-inflammatory cytokine gene expression in heart (IL-1$\beta$), pre-frontal cortex (IL-1$\beta$), and hypothalamus (IL-1$\beta$). The results provide mixed evidence for pro- and anti-inflammatory functions of corticosterone to regulate inflammatory responses to acute sleep loss.

Corresponding author
Zim Warda Hasan,
zimwarda.hasan200@topper.wku.edu

## INTRODUCTION

Sleep is considered a crucial mediator of the immune system and has a significant regulatory effect on immunological activities, enabling the host to resist infection and inflammation (*Shearer et al., 2005*). Sleep loss decreases the immune response and is linked to alterations in innate and adaptive immunity (*Korin et al., 2020*), increasing the risk of bacterial (*Everson, 1993*; *Everson & Toth, 2000*), viral, and protozoal diseases (*Friese, Bruns & Sinton, 2009*; *Lungato et al., 2015*). On the other hand, the immunological response, which is induced by an infection, affects sleep (*Foster et al., 2012*) resulting in the establishment of a bidirectional link between sleep and the immune system (*Besedovsky, Lange & Born, 2012*; *Besedovsky, Lange & Haack, 2019*).

Numerous factors, including lifestyle factors such as being overweight or sleep disorders (*e.g.*, insomnia, obstructive sleep apnea, and neurodegenerative disorders) can disrupt sleep (*Ogilvie & Patel, 2017*). Lack of sleep can influence glucose tolerance (*Chaput et al., 2007*), insulin sensitivity (*Nedeltcheva, Imperial & Penev, 2012*), formation of atherosclerotic plaques (*Cherubini et al., 2021*), oxidative stress (*Xue et al., 2019*), and the autonomic regulation of the brain (*Bobić et al., 2016*). These factors can lead to an increased risk of cardiovascular disorders such as coronary artery disease (*Aggarwal et al., 2013*), obesity (*Liu et al., 2013*), arrhythmias, diabetes mellitus (*Spiegel, Leproult & Van Cauter, 1999*), and hypertension (*Fang et al., 2012*; *Gottlieb et al., 2006*).

Mechanistically, sleep loss or dysfunction increases C-reactive protein, interleukin (IL)-6, and tumor necrosis factor (TNF) (*Irwin, Carrillo & Olmstead, 2010*), all of which are induced by NF-kappa-B (*Irwin et al., 2008*), which provides a link between increased inflammatory responses to many of the disorders described above. Pro-inflammatory cytokines are involved in the alteration of the neuroendocrine system and play an important role in sleep-wake cycling (*Krueger & Majde, 1995*). There is also an increase in pro-inflammatory responses in brain and peripheral tissues as a result of various types of sleep dysfunction (*e.g.*, sleep deprivation (*Chennaoui et al., 2015*), sleep restriction (*Yamanishi et al., 2022*), and sleep fragmentation (*Dumaine & Ashley, 2015*)). For example, previous research in our laboratory indicates that mice exposed to sleep fragmentation exhibit increased pro-inflammatory gene expression in brain and peripheral organs (*Dumaine & Ashley, 2015*; *Mishra et al., 2020*), and that this effect is rapid occurring within hours of onset of sleep fragmentation (*Nguyen, Fields & Ashley, 2023*).

Perturbations in sleep often lead to the stress response being activated, as measured by an activation of the hypothalamic-pituitary-adrenal (HPA) axis (*Kapsimalis et al., 2005*). Specifically, the paraventricular nucleus of the hypothalamus, the anterior lobe of the pituitary gland, and the adrenal gland are the main components of the HPA axis (*Nollet, Wisden & Franks, 2020*). Sleep disruption enhances the activity of the HPA axis and leads the hypothalamus to secrete corticotropin-releasing hormone (CRH) (*Cooper, Mishra & Ashley, 2019*; *Mongrain et al., 2010*). Adrenocorticotropic hormone (ACTH) is released by the anterior pituitary in response to CRH and acts on the adrenal cortex to stimulate secretion of glucocorticoids (GCs; cortisol in humans, corticosterone in mice) (*Chapotot et al., 2001*; *Wheeler et al., 2021*). Chronic exposure of GCs act as an anti-inflammatory

mediator, reducing the production of proinflammatory cytokines such as TNF-$\alpha$ and IL-1$\beta$ (*De Bosscher, Van den Berghe & Haegeman, 2003*). While GCs are well-known for their anti-inflammatory properties, in acute stress conditions, they can have a "priming" effect, shifting GCs towards pro-inflammatory effects (*Sorrells et al., 2009*). In the early stages of the stress response, GCs can elevate pro-inflammatory cytokines (TNF-$\alpha$ and IL-1$\beta$) in mice (*Johnson et al., 2002*; *Smyth et al., 2004*). Thus, this correlation between elevated pro-inflammatory markers and glucocorticoids requires further investigation, as it is unclear whether these steroid hormones are acting in a pro- or anti-inflammatory manner. Previous lab findings indicate that glucocorticoid administration during acute sleep fragmentation (ASF; 24 h) has a suppressive effect on proinflammatory cytokines in select tissues, although the dose of glucocorticoids used was supra-physiological (*Weaver, 2022*). Therefore, it is necessary to evaluate the role of glucocorticoids in mediating pro-inflammatory responses using alternative pharmacological methods.

In this study, we investigated the influence of GCs by pharmacological inhibition of either GC synthesis (metyrapone) or blockade of the GC receptor (mifepristone) upon mice exposed to acute sleep fragmentation (ASF; 24h) or control conditions (allowed to sleep freely). Metyrapone, a glucocorticoid synthesis inhibitor, acts by inhibiting the enzyme 11-$\beta$ hydroxylase, which catalyzes the conversion of 11-deoxycorticosterone to cortisol/corticosterone (*Roozendaal, Bohus & McGaugh, 1996*). Conversely, mifepristone inhibits the function of GC by blocking GC receptors, thereby inducing an elevation of the circulating cortisol (*Karena et al., 2022*) due to lack of negative feedback. Some studies have demonstrated that metyrapone successfully reduces the synthesis of GCs during SF (*Fernandes et al., 2020*; *Machado, Tufik & Suchecki, 2013*; *Raven et al., 2020*; *Tiba et al., 2008*). Similarly, mifepristone has been associated with the elevation of increased serum corticosterone in SF (*Ajibare, Ayodele & Olayaki, 2020*; *Demiralay et al., 2014*; *Wiedemann et al., 1992*) due to inhibition of negative feedback to the HPA axis. Nevertheless, the response of proinflammatory cytokines to GC synthesis inhibitors (metyrapone) or GC receptor antagonists (mifepristone) within the context of ASF has yet to be explored.

The specific goal of this study was to determine whether pharmacological inhibition of glucocorticoid synthesis and/or blockade of glucocorticoid receptors alters proinflammatory cytokines gene expression following 24 h of ASF in mice. If GCs can elevate pro-inflammatory cytokines in the early stage of stress response such as ASF, then we hypothesized that blocking the action of glucocorticoids by either inhibiting the synthesis of glucocorticoids using metyrapone or blocking glucocorticoid receptors using mifepristone would diminish the effect of proinflammatory responses to ASF compared with vehicle treatment. Furthermore, we predicted that blocking the action of glucocorticoid receptors using mifepristone, would have a greater effect upon pro-inflammatory cytokines in ASF than metyrapone, which reduces (but does not abolish) glucocorticoid synthesis. Alternatively, an elevation in proinflammatory responses from either of these drugs would indicate an anti-inflammatory effect of GCs on inflammatory responses.

## MATERIALS & METHODS

### Animals

For the experiment, male adult C57BL/6J mice ($n = 80$) were purchased from Jackson Laboratory and bred from the mouse colony housed in a mouse colony room at Western Kentucky University under standard rodent colony conditions (12:12-h light-dark cycle, lights on at 0800, 21 °C ± 1 °C). The selection of male adult C57BL/6J mice was based on their accessibility within our mouse colony. Throughout the study, mice were provided rodent chow (Rodent RM4 1800 diet (18% protein, 5% fat, 4% fiber), Cincinnati Lab Supply, Cincinnati, OH, USA) and tap water *ad libitum*. After weaning at 21 days of age, mice were placed into polypropylene cages comprised of same-sex littermates. The polypropylene mouse cages were provided with a thin layer of corncob bedding (Combo Bed-o-cobs, Cincinnati Lab Supply Inc., Cincinnati, OH, USA) and enrichment (Enrich-n' nest paper blend, Cincinnati Lab Supply Inc., Cincinnati, OH, USA). This research project was carried out under the approval of the Institutional Animal Care and Use Committee (IACUC) at Western Kentucky University (#22-07) and the procedures were performed following the National Institutes of Health's ''Guide for the Use and Care of Laboratory Animals'' and international ethics standards.

### Experiment design

Male mice (>8 weeks of age) were selected for the experiment and placed in an automated sleep fragmentation chamber to initiate sleep fragmentation (two per cage; SF) (Lafayette Instrument Company; Lafayette, IN, USA; model 80390). This device utilizes a standardized mouse cage with ad libitum food and water that has a horizontal bar installed that periodically moves across the floor of the cage at predetermined intervals, inducing SF but not absolute sleep deprivation (*Kaushal, Ramesh & Gozal, 2012*; *Dumaine & Ashley, 2015*). Mice were acclimated to the automated sleep fragmentation chambers for 72 h before the commencement of SF experiments to reduce any off-target effects from the unfamiliar cage setting (*Ashley et al., 2016*). Eighty mice were randomly assigned to groups, with sample sizes of 10 mice per group and only the first author was made aware of the randomization. Groups were divided into either ASF or control (non-sleep fragmentation; NSF) for each of the four different injection groups (see below).

### Experiment: acute sleep fragmentation

Following acclimation of 72 h, mice were anesthetized using isoflurane (5%) vapors (<2 min) to reduce any pain and distress and received one of the following treatments by subcutaneous injection: 0.9% NaCl ($n =20$; NSF ($n =10$), ASF ($n =10$)), 2% ethanol in polyethylene glycol ($n =20$; NSF ($n =10$), ASF ($n =10$)), metyrapone (a glucocorticoid synthesis inhibitor, 100 mg/kg BW; dissolved in 0.9% NaCl, $n =20$; NSF ($n =10$), ASF ($n =10$)), or mifepristone (a glucocorticoid receptor antagonist, 50 mg/kg BW; dissolved in 2% ethanol in polyethylene glycol, $n =20$; NSF ($n =10$), ASF ($n =10$)) at 8:20 am prior to the beginning of SF. The 0.9% NaCl and 2% ethanol in polyethylene glycol injections were used as vehicles for metyrapone and mifepristone, respectively (*Shearer et al., 2005*; *Smith-Swintosky et al., 1996*). Dosages of metyrapone and mifepristone were based on

previous studies that effectively inhibited the signaling of glucocorticoids in mice (*Blundell et al., 2011*; *Douma et al., 1998*; *Mesripour, Hajhashemi & Rabbani, 2008*; *Smith-Swintosky et al., 1996*). Ten minutes following injections (8:30 am, lights on), mice were subjected to ASF or control conditions (NSF) for 24 h. For ASF, the sweeping bar was programmed to move horizontally for 24 h at an interval of 120 s, or 30 arousals each hour. This rate is comparable to the sleep disturbances seen in people suffering from severe sleep apnea (*Goyal & Johnson, 2017*). The bar moves across the cage in 9 s, permitting enough time for mice to step over the bar to awaken them. There were no bar sweeps conducted on control mice, but they were housed in a sleep fragmentation chamber.

## Sample collection

After 24 h of ASF or NSF, mice from each group were rapidly anesthetized using isoflurane (5%) vapors (<2 min) and decapitated within 3 min of initial handling for tissue collection. For gene expression studies, three brain regions (prefrontal cortex (PFC), hypothalamus, and hippocampus) and four peripheral tissues (liver, spleen, heart, and epididymal white adipose tissue (EWAT)) were dissected from each of the 80 mice. For peripheral organs, subsamples of liver and EWAT were collected. Spleen and heart were collected whole. A total of 560 samples from both brain and peripheral regions were collected from 80 mice and preserved in RNAlater solution (Thermo Fisher Scientific, Waltham, MA, USA) at −20 °C for gene expression analysis. The dissection of brain regions followed the methods described by *Meyerhoff et al. (2021)*. These specific brain regions and peripheral tissues were selected because prior research indicated increased pro-inflammatory gene expression resulting from ASF (*Mishra et al., 2022*).

## RNA extraction

RNA was extracted from liver, spleen, EWAT, PFC, hypothalamus, and hippocampus using RNeasy mini kits (Qiagen, Hilden, Germany). NanoDrop 2000 Spectrophotometer (Thermo Fisher Scientific, Waltham, MA, USA) was used to determine RNA concentrations. RNA was isolated from the heart using a RNeasy fibrous tissue mini kit. All extractions were performed following the manufacturer's instructions and were performed on <30 mg/sample (these samples were selected randomly from the larger tissue sample).

## Reverse transcription and real time-PCR

Total RNA concentration of each tissue was diluted to the same concentration, and reverse transcribed into cDNA using a high-capacity cDNA reverse transcription kit (Thermo Fisher Scientific, Waltham, MA, USA, Cat number:4368813) according to the manufacturer's instructions. The prepared cDNA was used as a template to evaluate relative cytokine gene expression levels using an ABI 7300 RT-PCR system. Taq-Man gene expression RT-PCR master mix and the following cytokine probes: TNF-$\alpha$ (Mm00443258_m1), IL-1$\beta$ (Mm00434228_m1) were used. All probes were labeled with fluorescent marker 5-FAM at the 5′ end and quencher MGB at the 3′ end. The 18s endogenous control (primer-limited, VIC-labelled probe) was used according to the manufacturer's instructions. Samples were run in duplicates and the relative mRNA expression levels ($2^{-\Delta\Delta Ct}$) were obtained by measuring the fold change in mRNA level.

## Gene expression analysis

The cycle threshold (Ct) was used at which the fluorescence exceeded background levels to calculate $\Delta$Ct (Ct (target gene)–Ct (18S)). Each Ct value was normalized against the highest Ct value of a control sample ($\Delta$ $\Delta$Ct), and then the negative value of this powered to 2 ($2^{-\Delta\Delta Ct}$) was calculated for analysis.

## Statistical analysis

All statistical analyses were conducted using R Studio (v.1.3.1073, R Development Core Team, Boston, MA, USA). Animals and data points were excluded from the analysis if the Real-Time PCR readings were undetermined. The cycle threshold (Ct) values were used as criteria. If the Ct value was not detected within a set number of cycles, then the run was considered undetermined. Data were presented as mean ($\pm$SE) and $p < 0.05$ was considered statistically significant. Considering the multifactorial design of the research, a two-way ANOVA was applied to assess the influence of ASF and drug treatment as main factors and the interaction of these two factors. The relative gene expression of each proinflammatory cytokine (IL-1$\beta$, TNF-$\alpha$) was the dependent variable. Using the Shapiro–Wilk test and Levene's test, respectively, data groups were evaluated for normality and homogeneity of variances. *Post-hoc* analyses were conducted using Tukey's HSD test to identify specific differences between groups after assessing the main effects and interactions. In some cases, a logarithmic transformation was performed to fulfill the assumption of ANOVA. Nonparametric tests (Kruskal-Wallace H tests) were used when the assumption of ANOVA could not be met. *Post-hoc* analyses using Mann–Whitney U-tests were employed to assess differences between experimental groups for the latter. The Benjamini–Hochberg procedure was used to adjust for the multiple comparisons made.

## RESULTS

### Metyrapone experiment
#### *Peripheral response*

*Liver:* There were no significant effects of ASF, metyrapone, or their interaction on TNF-$\alpha$ (two-way ANOVA; ASF: $F_{1,36} = 0.02$; $p = 0.90$; metyrapone: $F_{1,36} = 0.07$; $p = 0.79$; interaction: $F_{1,36} = 0.19$; $p = 0.67$; Fig. 1A) or IL-1$\beta$ (two-way ANOVA; ASF: $F_{1,36} = 0.02$; $p = 0.88$, metyrapone: $F_{1,36} = 0.001$; $p = 0.97$; interaction: $F_{1,36} = 0.01$; $p = 0.91$; Fig. 1B) gene expression in liver.

*Spleen:* IL-1$\beta$ gene expression in spleen was significantly affected by ASF and metyrapone (Kruskal-Wallis; H (3) = 8.92; $p = 0.03$; Fig. 1D). However, there were no significant differences among groups using Mann Whitney U test (Mann Whitney; p >0.05). There were no significant differences between groups for TNF-$\alpha$ (Kruskal-Wallis; H (3) = 1.29; $p = 0.73$; Fig. 1C) gene expression.

*Epididymal white adipose tissue (EWAT):* There was a significant interaction between ASF and metyrapone upon TNF-$\alpha$ expression (two-way ANOVA; log-transformed; $F_{1,36} = 4.45$; $p = 0.04$; Fig. 1E). Tukey's *post hoc* tests indicated NSF-metyrapone mice exhibited elevated TNF-$\alpha$ gene expression compared with other groups (Tukey's HSD; p <0.05). Moreover,

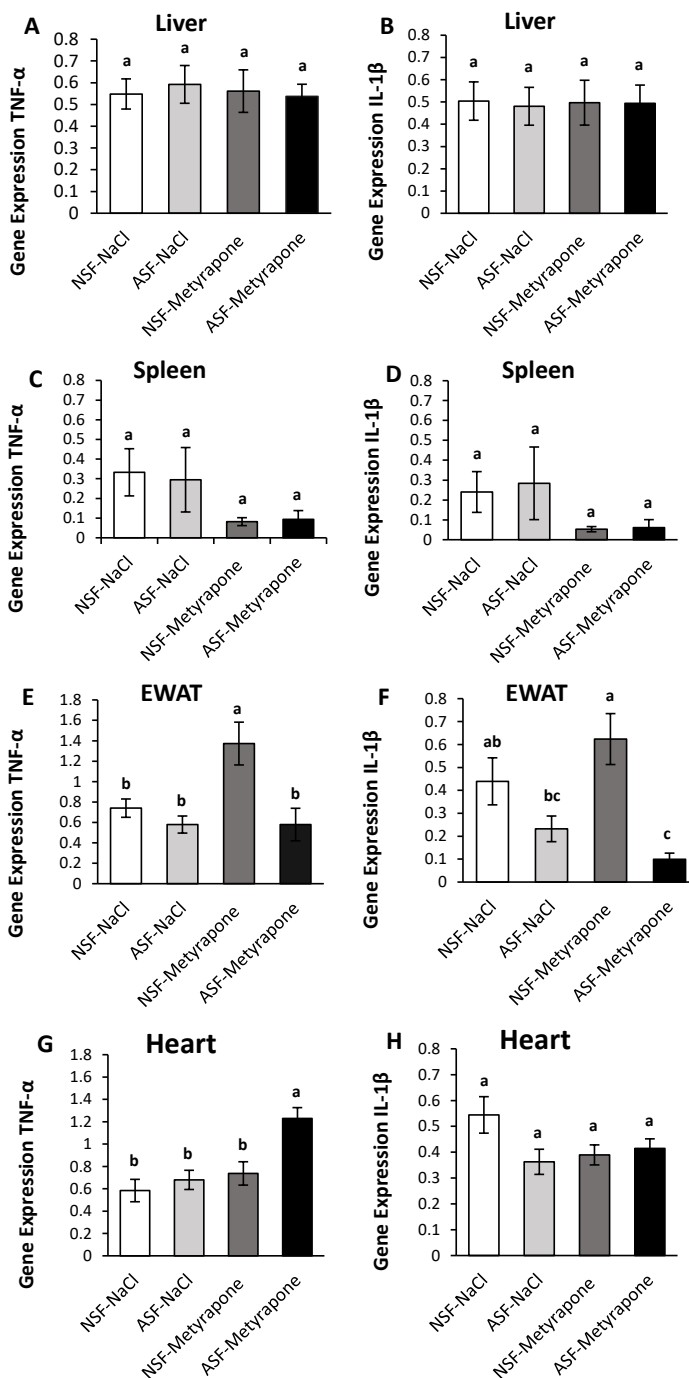

**Figure 1 Effects of acute sleep fragmentation (ASF), metyrapone, and their interaction upon cytokine gene expression in peripheral tissues.** Gene expression levels of TNF-$\alpha$ and IL-1$\beta$ were measured in liver (A, B), spleen (C, D), EWAT (E, F) and heart (G, H) of mice injected with vehicle (0.9% NaCl) or treated with metyrapone (glucocorticoid synthesis inhibitor) after being subjected to control or ASF. Data are presented as mean $\pm$ standard error (SE) for each group. The sample size of each group was $n = 10$ and data were analyzed either parametrically using a two-way ANOVA, followed by Tukey's HSD *post hoc* tests or non-parametrically using Kruskal-Wallis, followed by Mann Whitney U-tests. Significant differences were denoted by different letters among groups. Shared letters indicate no significant difference between groups. The level of statistical significance was set at alpha ($\alpha$) = 0.05.

there was a significant effect of ASF upon TNF-$\alpha$ expression in EWAT (two-way ANOVA; log-transformed; $F_{1,36} = 11.49$; $p = 0.002$; Fig. 1E). IL-1$\beta$ expression (Kruskal-Wallis; H (3) = 14.017; $p = 0.003$; Fig. 1F) revealed a significant difference among groups. Mann Whitney U tests indicated NSF-metyrapone mice exhibited elevated IL-1$\beta$ gene expression compared with other groups (Mann Whitney; p <0.05). Conversely, ASF-metyrapone mice exhibited significantly decreased IL-1$\beta$ gene expression compared with other groups (Mann Whitney; p <0.05).

*Heart:* In cardiac tissue, there was a significant interaction between ASF and metyrapone treatment upon TNF-$\alpha$ (two-way ANOVA; $F_{1,36} = 4.17$; $p = 0.049$; Fig. 1G) and IL-1$\beta$ (two-way ANOVA; $F_{1,36} = 4.18$; $p = 0.048$; Fig. 1H) expression. *Post hoc* Tukey's tests indicated that the ASF-metyrapone group exhibited increased TNF-$\alpha$ expression relative to other groups (Tukey's HSD; p <0.05). However, *post hoc* tests did not show any significant differences in IL-1$\beta$ gene expression among groups (Tukey's HSD; p >0.05). ASF and metyrapone also independently affected TNF-$\alpha$ expression in cardiac tissue (two-way ANOVA; ASF: $F_{1,36} = 9.14$; $p = 0.005$; metyrapone: $F_{1,36} = 13.09$; $p = 0.0009$; Fig. 1G).

### Brain response
*Prefrontal cortex (PFC):* There were no significant effects of ASF, metyrapone, or their interaction on TNF-$\alpha$ (two-way ANOVA; ASF: $F_{1,36} = 0.57$; $p = 0.46$; metyrapone: $F_{1,36} = 0.07$; $p = 0.80$; interaction: $F_{1,36} = 0.43$; $p = 0.52$; Fig. 2A) or IL-1$\beta$ (two-way ANOVA; log-transformed; ASF: $F_{1,36} = 0.11$; $p = 0.74$, metyrapone: $F_{1,36} = 0.009$; $p = 0.93$; interaction: $F_{1,36} = 0.18$; $p = 0.67$; Fig. 2B) gene expression in prefrontal cortex.

*Hypothalamus:* In the hypothalamus, there was a significant effect of ASF (two-way ANOVA; log-transformed; ASF: $F_{1,36} = 7.85$; $p = 0.008$; Fig. 2C), but no effects from metyrapone or their interaction (two-way ANOVA; log-transformed; metyrapone: $F_{1,36} = 0.07$; $p = 0.80$; interaction: $F_{1,36} = 3.70$; $p = 0.06$; Fig. 2C). No significant differences were detected among groups for IL-1$\beta$ (Kruskal-Wallis; H (3) = 2.07; $p = 0.56$; Fig. 2D) expression.

*Hippocampus:* There were no significant effects of ASF, metyrapone, or their interaction on TNF-$\alpha$ (two-way ANOVA; ASF: $F_{1,36} = 0.04$; $p = 0.84$; metyrapone: $F_{1,36} = 0.01$; $p = 0.92$; interaction: $F_{1,36} = 0.37$; $p = 0.55$; Fig. 2E) or IL-1$\beta$ (two-way ANOVA; ASF: $F_{1,36} = 0.99$; $p = 0.32$, metyrapone: $F_{1,36} = 3.14$; $p = 0.09$; interaction: $F_{1,36} = 0.03$; $p = 0.85$; Fig. 2F) gene expression in hippocampus.

## Mifepristone experiment
### Peripheral response
*Liver:* In hepatic tissue, there was a significant difference among groups for TNF-$\alpha$ gene expression (Kruskal-Wallis; H (3) = 10.40; $p = 0.02$; Fig. 3A). A Mann Whitney U test revealed decreased TNF-$\alpha$ gene expression in NSF-mifepristone mice compared with vehicle (Mann Whitney; p <0.05). There were no significant effects detected among groups for IL-1$\beta$ (Kruskal-Wallis; H (3) = 6.53; $p = 0.09$; Fig. 3B) gene expression.

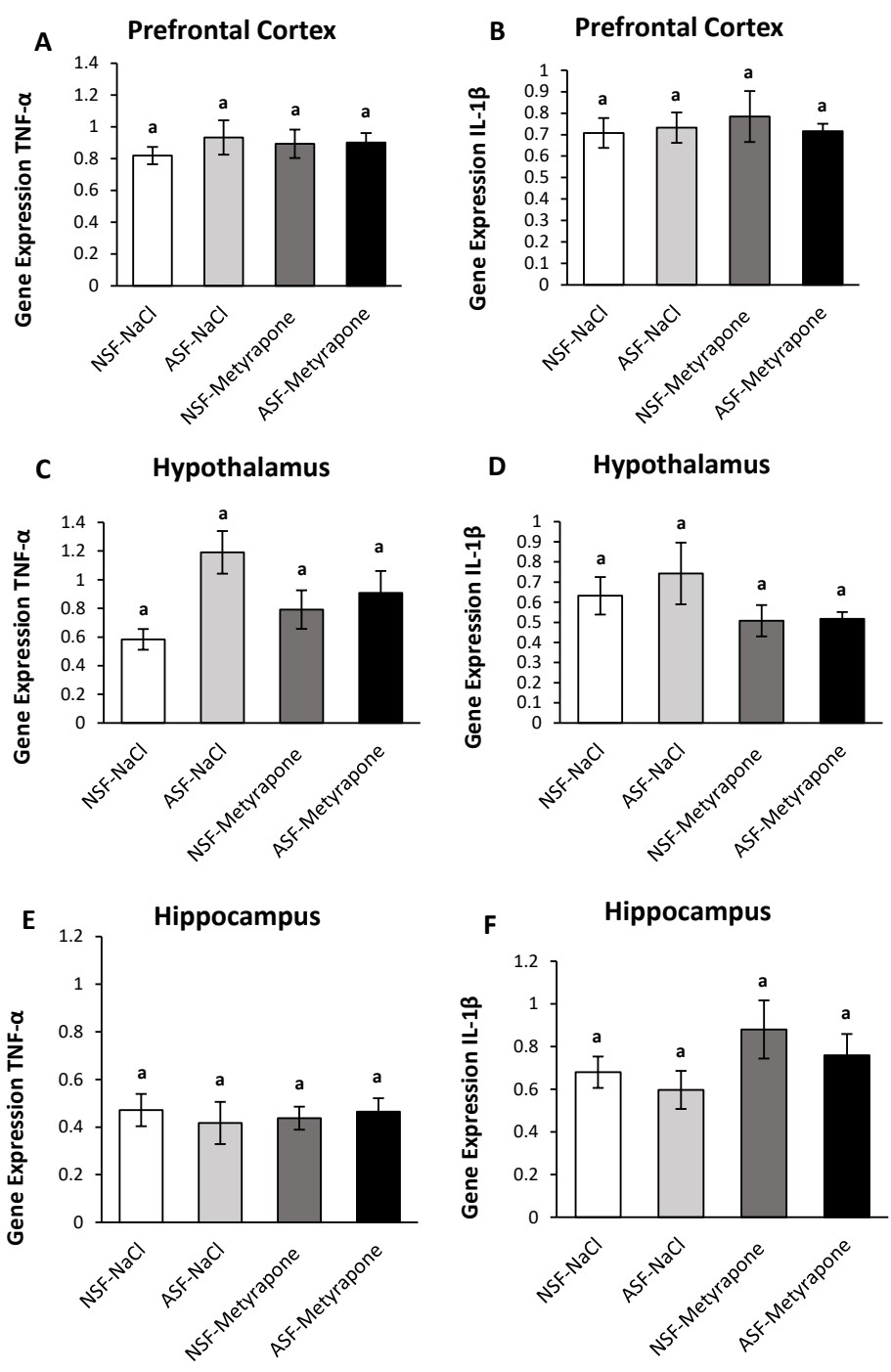

**Figure 2  Effects of acute sleep fragmentation (ASF), metyrapone, and their interaction upon cytokine gene expression in select regions of brain.** Gene expression levels of TNF-$\alpha$ and IL-1$\beta$ were measured in PFC (A, B), hypothalamus (C, D), hippocampus (E, F) of mice injected with vehicle (0.9% NaCl) or treated with metyrapone (glucocorticoid synthesis inhibitor) after being subjected to control or ASF. Data are presented as mean ± standard error (SE) for each group. The sample size of each group was $n = 10$ and data were analyzed either parametrically using a two-way ANOVA, followed by Tukey's HSD *post hoc* tests or non-parametrically using Kruskal-Wallis, followed by Mann Whitney U-tests. Significant differences among groups denoted as by different letters. Shared letters indicate no significant difference between groups. The level of statistical significance was set at alpha ($\alpha$) = 0.05.

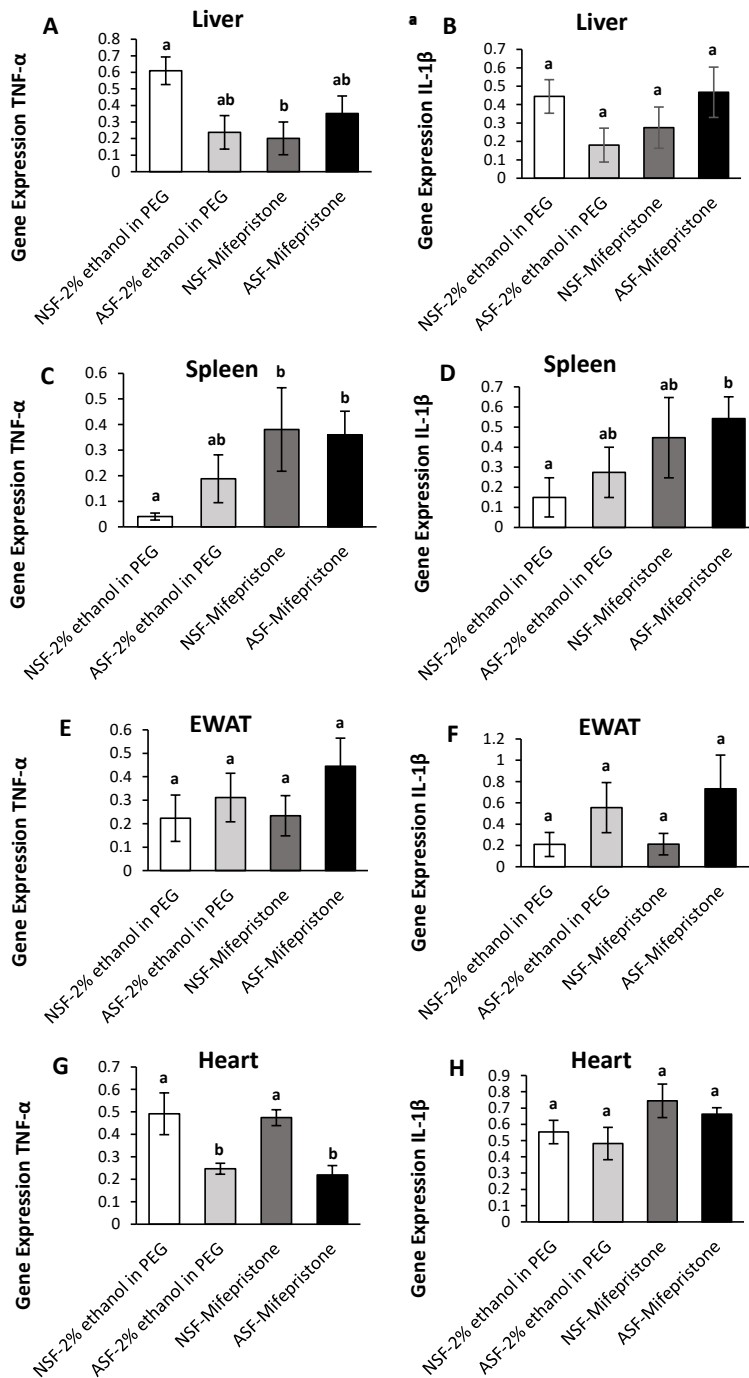

**Figure 3** **Effects of acute sleep fragmentation (ASF), mifepristone, and their interaction upon cytokine gene expression in peripheral tissues.** Gene expression levels of TNF-$\alpha$ and IL-1$\beta$ were measured in liver (A, B), spleen (C, D), EWAT (E, F) and heart (G, H) of mice injected with vehicle (2% ethanol in polyethylene glycol) or treated with mifepristone (glucocorticoid receptor antagonist) after being subjected to control or ASF. Data are presented as mean $\pm$ standard error (SE) for each group. The sample size of each group was $n = 10$ and data either parametrically using a two-way ANOVA, followed by Tukey's HSD *post hoc* tests or non-parametrically using Kruskal-Wallis, followed by Mann Whitney U-tests. Significant differences were denoted by different letters among groups. Shared letters indicate no significant difference between groups. The level of statistical significance was set at alpha $(\alpha) = 0.05$.

*Spleen:* There were significant effects among groups for TNF-$\alpha$ (Kruskal-Wallis; H (3) = 14.51; $p = 0.002$; Fig. 3C) and IL-1$\beta$ (Kruskal-Wallis; H (3) = 10; $p = 0.02$; Fig. 3D) gene expression in spleen. Mann Whitney U tests indicated that TNF-$\alpha$ gene expression was elevated in NSF-mifepristone and ASF-mifepristone relative to other groups, while IL-1$\beta$ gene expression was also elevated in ASF-mifepristone mice compared with other groups (Mann Whitney; p <0.05).

*Epididymal white adipose tissue (EWAT):* There was a significant difference among groups for IL-1$\beta$ (Kruskal-Wallis; H (3) = 8.3233; $p = 0.04$; Fig. 3F) gene expression. However, the Mann U Whitney tests did not detect any significant differences among groups (Mann Whitney; p >0.05). There were no significant differences detected in TNF-$\alpha$ expression in EWAT (Kruskal-Wallis; H (3) = 5.2756; $p = 0.15$; Fig. 3E).

*Heart:* There was a significant difference among groups for TNF-$\alpha$ (Kruskal-Wallis; H (3) = 17.32; $p = 0.0006$; Fig. 3G) gene expression in heart. Mann Whitney U tests revealed decreased TNF-$\alpha$ gene expression in ASF-vehicle and ASF-mifepristone mice compared with other groups (Mann Whitney; p <0.05). IL-1$\beta$ (Two -way ANOVA; mifepristone: $F_{1,36} = 5.09$; $p = 0.03$ Fig. 3H) was significantly affected by mifepristone treatment, but a significant interaction effect (Two -way ANOVA; $F_{1,36} = 0.005$; $p = 0.95$; Fig. 3H) was not detected.

### Brain response
*Prefrontal cortex (PFC):.* IL-1$\beta$ (two-way ANOVA; mifepristone: $F_{1,36} = 7.67$; $p = 0.009$; Fig. 4B) gene expression was significantly affected by mifepristone treatment in PFC. However, no interaction effect was observed between ASF and mifepristone treatment for TNF-$\alpha$ (two-way ANOVA; $F_{1,36} = 0.05$; $p = 0.82$; Fig. 4A) or IL-1$\beta$ (two-way ANOVA; log-transformed; $F_{1,36} = 1.39$; $p = 0.25$; Fig. 4B) gene expression.

*Hypothalamus:* There were significant effects of ASF, mifepristone, but not their interaction on IL-1$\beta$ (two-way ANOVA; log-transformed; ASF: $F_{1,36} = 6.95$; $p = 0.01$; mifepristone: $F_{1,36} = 12.85$; $p = 0.0009$; interaction; $F_{1,36} = 2.17$; $p = 0.15$; Fig. 4D) gene expression in hypothalamus. There were no significant effects of ASF, mifepristone, or their interaction on TNF-$\alpha$ (two-way ANOVA; ASF: $F_{1,36} = 0.41$; $p = 0.52$; mifepristone: $F_{1,36} = 3.22$; $p = 0.08$; interaction; $F_{1,36} = 0.05$; $p = 0.83$; Fig. 4C) gene expression.

*Hippocampus:* There was no significant effect of ASF or mifepristone on TNF-$\alpha$ (two-way ANOVA; ASF: $F_{1,36} = 3.15$; $p = 0.08$; mifepristone: $F_{1,36} = 0.03$; $p = 0.87$; Fig. 4E) expression but a significant interaction between ASF and mifepristone upon hippocampal TNF-$\alpha$ (two-way ANOVA; $F_{1,36} = 5.81$; $p = 0.02$; Fig. 4E) gene expression was detected. *Post hoc* Tukey's tests revealed increased TNF-$\alpha$ gene expression in ASF-vehicle mice relative to control (Tukey's HSD; p <0.05). There was no significant effect of ASF, mifepristone or their interaction on IL-1$\beta$ (two-way ANOVA; ASF: $F_{1,36} = 3.97$; $p = 0.05$, mifepristone: $F_{1,36} = 0.02$; $p = 0.89$; interaction: $F_{1,36} = 3.77$; $p = 0.06$; Fig. 4F) gene expression in hippocampus. The raw data for this study has been provided as Supplemental Files.

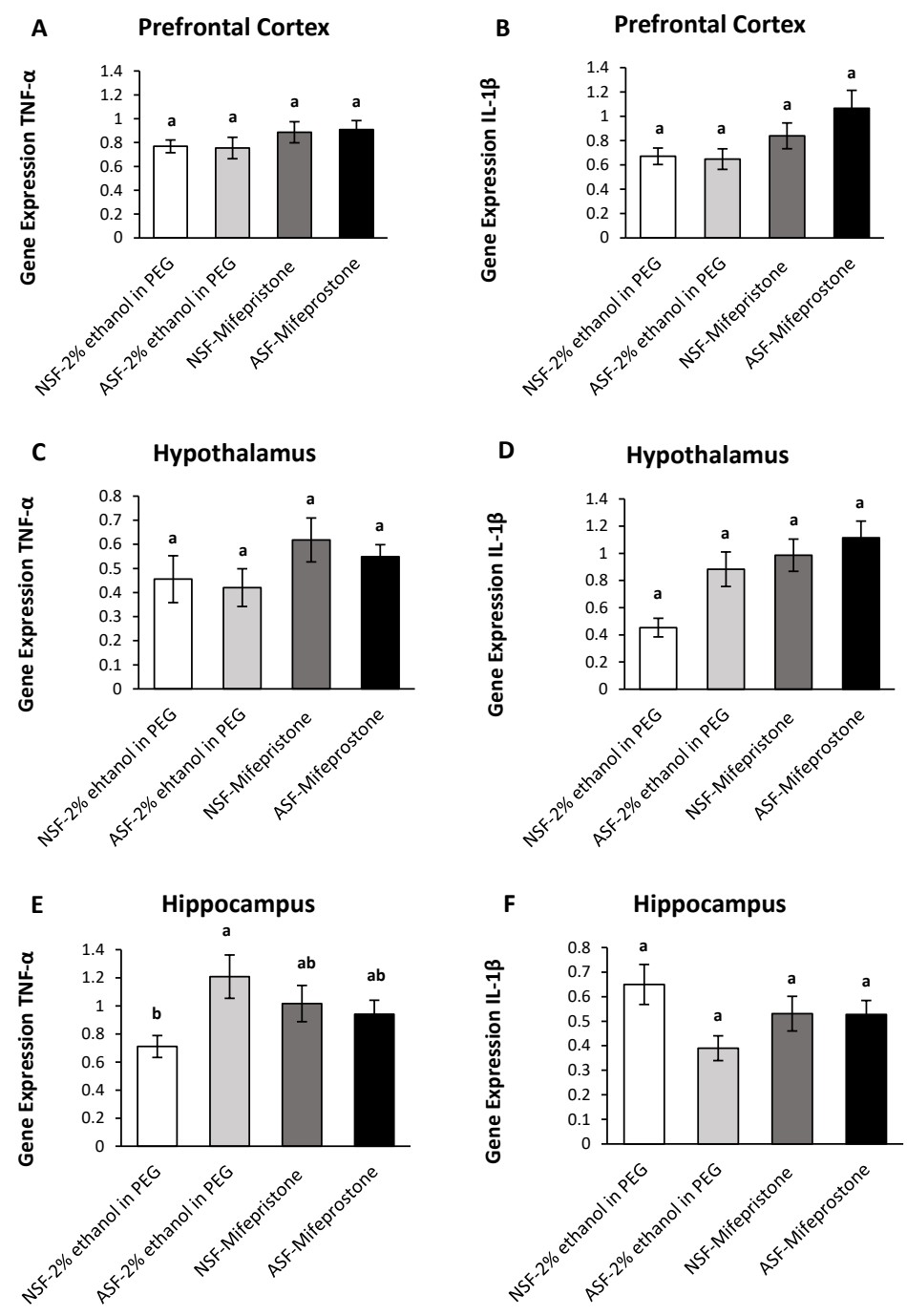

**Figure 4** **Effects of acute sleep fragmentation (ASF), mifepristone, and their interaction upon cytokine gene expression in select regions of brain.** Gene expression levels of TNF-$\alpha$ and IL-1$\beta$ were measured in PFC (A, B), hypothalamus (C, D), hippocampus (E, F) of mice injected with vehicle (2% ethanol in polyethylene glycol) or treated with mifepristone (glucocorticoid receptor antagonist) after being subjected to control or ASF. Data are presented as mean $\pm$ standard error (SE) for each group. The sample size of each group was $n = 10$ and data were analyzed using a two-way ANOVA, followed by Tukey's HSD *post hoc* tests or non-parametrically using Kruskal-Wallis, followed by Mann Whitney U-tests. Significant differences were observed between certain groups, as denoted by different letters. Shared letters indicate no significant difference between groups. The level of statistical significance was set at alpha $(\alpha) = 0.05$.

## DISCUSSION

Results of this research provide evidence that glucocorticoids have variable and complex effects upon pro-inflammatory cytokine expression among mice exposed to ASF, with some tissues producing pro-inflammatory actions and others an anti-inflammatory effect. In addition, mifepristone treatment, which blocks GC receptors, had a greater effect upon regulation of pro-inflammatory gene expression than metyrapone, which reduces GC production. Mice treated with mifepristone exhibited an interaction with ASF in four peripheral tissues: liver (TNF-$\alpha$), spleen (TNF-$\alpha$ and IL-1$\beta$), EWAT (IL-1$\beta$), heart (TNF-$\alpha$), as well as in brain, specifically hippocampal (TNF-$\alpha$) tissue. In contrast, mice treated with metyrapone only displayed an interaction with ASF in three peripheral tissues: spleen (IL-1$\beta$), EWAT (TNF-$\alpha$ and IL-1$\beta$) and cardiac tissue (TNF-$\alpha$ and IL-1$\beta$), and no brain tissue. These results are consistent with our hypothesis that blocking the action of glucocorticoid receptors (mifepristone) would have a greater effect upon pro-inflammatory cytokines during ASF compared with suppressing (but not eliminating) the synthesis of GCs (metyrapone).

Metyrapone treatment diminished pro-inflammatory gene expression in spleen (IL-1$\beta$) and adipose (IL-1$\beta$) tissue after ASF and in cardiac tissue (TNF-$\alpha$) from mifepristone treatment. This unexpected finding partially supports a possible pro-inflammatory effect of GCs on these tissues. However, elevated TNF-$\alpha$ expression in cardiac tissue following metyrapone treatment implies an anti-inflammatory effect of GCs. In comparison, mifepristone increased pro-inflammatory gene expression in heart (IL-1$\beta$), pre-frontal cortex (IL-1$\beta$) and hypothalamus (IL-1$\beta$), suggesting a broader influence of mifepristone than metyrapone and an overwhelming anti-inflammatory effect of GCs. As a result, this finding reveals that pro-inflammatory *vs.* anti-inflammatory responses among ASF mice are tissue-specific, especially in the periphery. Reasons for this discrepancy are unclear but could be due to the density of GC receptors at these different organs (*Basarrate et al., 2024*), the sensitivity of these receptors to reduced corticosterone *versus* complete receptor blockade, and the different clearance rates of these two drugs that could affect inflammatory responses at different time points of the 24 h of ASF (*Heikinheimo, 1989*).

In our study, elevated proinflammatory cytokines (TNF-$\alpha$) were detected in hypothalamus and cardiac tissue following ASF in the metyrapone study (Table 1), as well as elevated pro-inflammatory cytokines in hypothalamus (IL-1$\beta$) in the mifepristone study (Table 2). These findings are consistent with previous findings in our laboratory regarding increased pro-inflammatory gene expression in hypothalamus and cardiac tissue (*Nguyen, Fields & Ashley, 2023*) after ASF. Additionally, contrary to earlier findings, proinflammatory cytokines (TNF-$\alpha$) were suppressed in EWAT tissue rather than elevated (*Ensminger et al., 2022*) after treatment with metyrapone. Reasons for this discrepancy are unclear but may reflect differences in the pharmacodynamics and clearance rates of the two drugs. Metyrapone has a short half-life of around 1.9 h (*Kumari & Willing, 2022*). And the clearance rate of mifepristone is 1.5–6 L/h/kg in rats, although the bioavailability of mifepristone in humans is prolonged due to its binding to $\alpha$1 glycoprotein, resulting in a

**Table 1  Summary of effects from ASF, metyrapone and their interaction upon TNF-α and IL-1β gene expression in brain and peripheral tissues.**

| Tissue Type | Sleep Fragmentation | Metyrapone | Interaction Between ASF and Treatment |
|---|---|---|---|
| Liver | No effect | No effect | No effect |
| Spleen | No effect | No effect | TNF-α-No effect<br>IL-1β -Interaction between ASF and metyrapone but no significant differences among groups using post hoc tests. |
| EWAT | TNF-α- ↓<br>IL-1β—No effect | No effect | TNF-α-NSF Metyrapone ↑<br>IL-1β -NSF Metyrapone ↑<br>IL-1β -ASF Metyrapone ↓ |
| Heart | TNF-α- ↑<br>IL-1β—No effect | TNF-α- ↑<br>IL-1β—No effect | TNF-α-ASF Metyrapone ↑<br>IL-1β -Interaction between ASF and metyrapone but the post hoc tests did not reveal significant interaction between groups. |
| Prefrontal Cortex | No effect | No effect | No effect |
| Hypothalamus | TNF-α- ↑<br>IL-1β—No effect | No effect | No effect |
| Hippocampus | No effect | No effect | No effect |

**Notes.**

↑ denotes upregulation of gene expression, ↓ denotes downregulation of gene expression.

much longer half-life compared to rodents, who lack this binding protein (*Wulsin, Herman & Danzer, 2016*).

The effects of metyrapone were stronger in peripheral tissue than in brain (Table 1). Metyrapone readily crosses the blood–brain barrier (*Stith, Person & Dana, 1976*), thus this lack of an effect in brain is not due to limited bioavailability. Neurodegeneration in the hippocampus occurs from prolonged exposure to elevated GCs (*Landfield, 1987*). Consequently, chronic SF could potentially affect hippocampal function and play a role in the development of neurodegeneration (*Korin et al., 2020*; *Krugers et al., 1998*). Metyrapone has been found to have a protective effect against several neurological insults (*Sapolsky, Krey & McEwen, 1985*). Surprisingly, metyrapone had no effect upon proinflammatory cytokines (TNF-α, IL-1β) in brain (PFC, hypothalamus, hippocampus) gene expression among mice subjected to ASF or NSF. It is possible that the decreases in circulating GCs from metyrapone treatment were not sufficient enough to alter responses to GCs in brain, but further study is required. Another interesting aspect of metyrapone is its very short half-life (1.9 h). It is therefore possible that the effect of the drug 24 h later could produce no effect or even a compensatory effect. The specific timeframe during which GC transitions from a proinflammatory state to an anti-inflammatory state in response to stress circumstances may have a possible influence on these findings (*Duque & Munhoz, 2016*). Current research in our laboratory indicates that the HPA axis is activated rapidly (within 1 h) after the commencement of ASF and remains elevated for at least 24 h of ASF (*Nguyen, Fields & Ashley, 2023*). Interestingly, the first tissue to exhibit an increase in pro-inflammatory cytokine expression is the heart (1 h of ASF) despite serum corticosterone levels being elevated. Results of the mifepristone study suggest that GCs could potentially exert pro-inflammatory effects in heart, but this result conflicts with the

**Table 2** Summary of effects from ASF, mifepristone and their interaction upon TNF-α and IL-1β gene expression in brain and peripheral tissues.

| Tissue Type | Sleep Fragmentation | Mifepristone | Interaction Between ASF and Treatment |
|---|---|---|---|
| Liver | No effect | No effect | TNF-α-NSF—Mifepristone ↓<br>IL-1β-No effect |
| Spleen | No effect | No effect | TNF-α-ASF—Mifepristone ↑<br>TNF-α-NSF -Mifepristone ↑<br>IL-1β-ASF—Mifepristone ↑ |
| EWAT | No effect | No effect | TNF-α-No effect<br>IL-1β -Interaction between SF and metyrapone but the post hoc test did not reveal significant interaction between groups. |
| Heart | No effect | TNF-α-No effect<br>IL-1β- ↑ | TNF-α-ASF—vehicle ↓<br>TNF-α -ASF—Mifepristone ↓<br>IL-1β -No effect |
| Prefrontal Cortex | No effect | TNF-α-No effect<br>IL-1β- ↑ | No effect |
| Hypothalamus | TNF-α-No effect<br>IL-1β - ↑ | TNF-α-No effect<br>IL-1β- ↑ | No effect |
| Hippocampus | No effect | No effect | TNF-α-ASF—vehicle ↑<br>IL-1β-No effect |

**Notes.**

↑ denotes upregulation of gene expression, ↓ denotes downregulation of gene expression.

metyrapone study, where blockade of GC synthesis leads to an elevation in cardiac TNF-α gene expression 24 h after SF (Table 1).

Compared with metyrapone, mifepristone (which blocks GC receptors) had more potent effects across most tissues. Glucocorticoids act on tissues by binding to two receptors: the glucocorticoid receptor (GR) and the mineralocorticoid receptor (MR). In the brain, GRs are ubiquitously distributed, with a high concentration in the hypothalamus and pituitary, specifically in CRH neurons and pituitary corticotropes, respectively (*Ahima & Harlan, 1990*), whereas the MR is highly expressed in the limbic system, specifically the hippocampus and some peripheral tissues (*e.g.*, kidney, heart, colon) (*Funder, 2005*). The drug mifepristone is an antiprogestin and anticortocosteroid, but it may also affect mineralocorticoid receptors directly or indirectly (*Agarwai, 1996*). In hippocampus, ASF increased TNF-α gene expression compared with NSF-vehicle mice (Table 2). However, when treated with mifepristone, the effect from ASF was abolished. Conversely, in the metyrapone study (Table 1), ASF did not induce an elevation in TNF-α expression relative to NSF-vehicle mice in hippocampus. This discrepancy suggests the possibility that the mice could respond differently to different vehicles (0.9% NaCl and 2% ethanol in polyethylene glycol) during ASF. In other measured brain areas, mifepristone increased IL-1β expression (regardless of ASF), implying an anti-inflammatory role of GCs in brain. Furthermore, elevated proinflammatory cytokines were detected in hypothalamus (IL-1β) among ASF mice. This finding is consistent with previous research in our lab regarding hypothalamus (*Dumaine & Ashley, 2018*).

In peripheral tissues, there are some contrasting effects from mifepristone treatment. In spleen, mifepristone elevated IL-1β and TNF-α expression compared to vehicle mice

during ASF, emphasizing an anti-inflammatory role of GCs in splenic tissue (Table 2). A pro-inflammatory effect of GCs was detected (TNF-$\alpha$) in liver but was only seen among NSF mice. While an interaction between ASF and mifepristone (IL-1$\beta$) was observed in EWAT, subsequent *post-hoc* tests did not reveal any significant differences between groups. Moreover, in cardiac tissue, an interaction between ASF and mifepristone (TNF-$\alpha$) was found. *Post-hoc* analysis revealed a diminished TNF-$\alpha$ gene expression in ASF-vehicle and ASF- mifepristone mice compared to other groups. These findings support an anti-inflammatory role of GCs in spleen, but potentially a pro-inflammatory effect in liver (but only among NSF mice) and heart.

## CONCLUSIONS

Sleep disorders that cause sleep fragmentation, such as sleep apnea and insomnia, are on the rise, contributing to inflammation and chronic damage to other organs. In addition to that, increased cortisol-producing conditions such as hormonal disorders, prolonged steroid use, certain types of cancer and psychiatric disorders has been associated with sleep disturbances. Several studies reported a proinflammatory effect of GCs which can lead to initiation and propagation of inflammation (*Sorrells et al., 2009*). However, the relationship of ASF with the pro-inflammatory effect of GCs is not well established. We provided evidence that when exposed to metyrapone, GCs promote pro-inflammatory reactions in spleen and adipose tissue, whereas treatment with mifepristone induces similar reactions in the heart during ASF. We also found that among metyrapone-treated mice, GC action is anti-inflammatory in adipose tissue (NSF-metyrapone group only) and heart, and in the spleen when treated with mifepristone during ASF. These findings may relate to GC type and to the concentration of GC receptors in brain and peripheral organs as well as precise time frame during which GC will convert from proinflammatory to anti-inflammatory in the exposure of stress conditions. There is a possibility that the lack of consistency between the two studies might be due to the heterogeneity of tissue sampling methods. Future research should focus on determining the specific time course during which glucocorticoids change from a proinflammatory to an anti-inflammatory function under stress, as well as the impact on inflammation in organs during ASF. Furthermore, our study underscores the significance of extending these investigations to explore the effects of chronic sleep fragmentation, thereby providing a more comprehensive understanding of the effects of blocking glucocorticoid action on inflammatory responses. It is conceivable that the relationship between HPA activation and chronic sleep fragmentation deviates from the ASF findings reported in this study. Therefore, experiments that explore the effects of glucocorticoid action on inflammatory responses to chronic sleep fragmentation are warranted. Finally, mifepristone was more effective in regulating proinflammatory cytokines as it blocks GC receptors compared with metyrapone which will reduce the synthesis of glucocorticoids. These findings could lead to therapeutic approaches for treating sleep disorders and increased glucocorticoid secretion associated with sleep disturbances such as Cushing syndrome.

## ACKNOWLEDGEMENTS

We thank Naomi Rowland for assistance with RT-PCR and other protocols. We also thank Dr. Jarrett Johnson for assistance with statistical analysis.

### Funding

The authors received funding from the National Institutes of General Medical Sciences grants: R15GM117534-02 and P20GM103436-22 as well as a Western Kentucky University Graduate Studies Research Grant. The funders had no role in study design, data collection and analysis, decision to publish, or preparation of the manuscript.

### Grant Disclosures

The following grant information was disclosed by the authors:
The National Institutes of General Medical Sciences grants: R15GM117534-02, P20GM103436-22.

### Competing Interests

The authors declare there are no competing interests.

### Author Contributions

- Zim Warda Hasan performed the experiments, analyzed the data, prepared figures and/or tables, authored or reviewed drafts of the article, and approved the final draft.
- Van Thuan Nguyen performed the experiments, authored or reviewed drafts of the article, and approved the final draft.
- Noah T. Ashley conceived and designed the experiments, analyzed the data, authored or reviewed drafts of the article, and approved the final draft.

### Animal Ethics

The following information was supplied relating to ethical approvals (*i.e.*, approving body and any reference numbers):

Institutional Animal Care and Use Committee (IACUC) at Western Kentucky University

### Data Availability

The raw measurements are available in the Supplementary Files.

### Supplemental Information

Supplemental information for this article can be found online at http://dx.doi.org/10.7717/peerj.17539#supplemental-information.

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
