# Peer review of "Effect of glucocorticoid blockade on inflammatory responses to acute sleep fragmentation in male mice"

_PeerJ, doi:10.7717/peerj.17539_

## Round 0.1 · original submission · Major Revisions

Dear Authors:

Thank you for considering PeerJ to submit your manuscript titled: "Effect of glucocorticoid blockade on inflammatory responses to acute sleep fragmentation in mice." After an exhaustive review by two anonymous reviewers and myself, I consider that the document requires substantial improvements to adjust to our journal's quality standards. Could you please take the reviewers' comments into account?

Cordially,

Dr. Manuel Jimenez

·

Excellent Review

This review has been rated excellent by staff (in the top 15% of reviews)
EDITOR COMMENT
Thank you very much for your excellent review. Many of the issues you discuss particularly concerned me after first reading the body of the manuscript. Without a doubt, the review was exhaustive and detailed. Reviews of this quality are well received at PeerJ because its ultimate goal is always to improve the quality of the final manuscript. Thank you very much for your effort and good work; it is a pleasure and an honor to receive reviews of this quality and commitment to science.

Basic reporting

• I encountered labeling errors in my review (misplaced sentences involving statistical tests, misnumbered tables, and discordance between results text and tables), which suggests that overall, the authors should carefully review all text, figures, and tables for consistency and accuracy.
• Could the authors discuss the potential off-target effects of the drugs used in this study and how these could explain some of their results. For example, activity at MR receptors is affected by metyrapone via build-up of precursor steroids, and activity at MRs has its own complex relationship with inflammation in various tissues.
• For some of the figure panels, letters denoting significant differences are used. Whereas, in others, nothing is used, whether there are significant findings or not. Please edit so the depictions of the findings are consistent within a figure.
• Line 118: Cytokines
• Line 134: chaw -> chow
• Line 165: the time has a period in place of a colon.
• Line 184: also stored at -20 C?
• Lines 375: Define ‘Cort’ as Corticosterone.
• Line 412: Correct to “blockade of GC synthesis”
• Line 416: Correct to “In the brain”.
• Paragraph beginning on line 414: It’s not clear what the authors mean when discussing the different GC receptor types, including discussion of the thalamus, which wasn’t looked at in this study. Are they implying that GC type I is most prevalent in peripheral tissues, and that type II is more prevalent in the brain areas they quantified?
• Line 770: Tables 1 and 2 are misnumbered compared to the order they later appear in.
• Table 1 appears before figure 4, but would make much more sense if included after figure 4.
• In the table captions, please indicate what the arrows represent. Is it always representing a change with respect to the NSF control group? I assume that sometimes the differences are with some groups but not all other groups.
• In the tables:
o In some cases “No effect” is included for a gene, but in other cells there is no mention of the gene when there is no effect.
o In cases where there are multiple comparisons highlighted in the interactions column, it’s not clear that the second comparison is still related to the same gene, or whether it’s just not labeled at all (e.g., Mifepristone table, interaction column, Spleen row, text reading “NSF- Mifepristone ^”).
o For the main effect columns, because the columns already have the name of the treatment, it’s redundant to also include those treatment names in each cell.

Experimental design

• Two mice were included per sleep deprivation cage, so these replicates are not independent. The appropriate analysis would be to account for this using “cage” as a random effect in a linear mixed model.
• The methods around sample collection don’t have enough detail to replicate the study.
o Please include a schematic of where central and peripheral tissue samples were taken from. Including: What was the size of the sample? What method was used for consistency across samples? For peripheral tissues, I am assuming that samples were taken consistently from a part of the organ (i.e., what part of the heart was sampled). Or, were the whole organs ground up for RNA extraction?
• Line 213: please elaborate on what specific criteria were used to determine when a run was undetermined.
• Line 215: Add, “and the interaction of these two factors”, as this was the main hypothesis of interest in the study.
• Line 221: I suggest tweaking wording to clarify that you used non-parametric tests in the case that ANOVA assumptions could not be met. Then in the line below, clarify that Mann-Whitney U tests were used as post hoc analysis.

Validity of the findings

• The authors focus exclusively on male mice, though the scope of inference isn't made clear. Please add a short justification for the choice to use only males. The end of the title of the manuscript should be changed to “…in male mice”.
• Something that stuck out to me is that the authors perform the same experiment in two cohorts of mice (two different experiments comparing NSF-vehicle to ASF-vehicle) and sometimes see different findings. As examples, for TNF-a expression in the hypothalamus, there’s an elevation in the metyrapone study, but not in the mifepristone study…whereas in hippocampus, there’s an elevation in the mifepristone study but not in the metyrapone study.
o Do the authors attribute this to the vehicle itself?
o It’s hard to imagine the ASF paradigm itself would lead to such different results in the two cohorts, unless the authors know of something that happened in the animal rooms?
o Another possibility I could imagine is variability in what part of an organ was sampled in one cohort compared to another.
o The authors should discuss how for some tissue/region combinations, a lack of consistency between the two studies could drive some of the interactions and effects of interest. This has implications for the results that address their main hypothesis, as well as their discussion about the two drugs differing in their effectiveness.
• Overall, I found the results were difficult to parse as written, so some of my concerns may be simple fixes by carefully revising the text.
o In many cases, Tukey statistics are not reported inline, but should be, including the statistics (or added in a separate table).
o I became unclear how post-hoc tests were being run as I went through the results. Were Mann-Whitney U tests run between all groups following a significant Kruskal Wallace test? I only ever see two tests, but there are 4 experimental groups. This is particularly important because the authors’ main hypotheses have to do specifically with the interaction of drug and sleep treatments, which couldn’t be tested with only 2 Mann-Whitney tests.
o The Tukey tests are protected for multiple comparisons, but the Mann-Whitney U tests are not. The authors should use some type of correction, especially since non-parametric tests account for a little under 50% of their tests.
o As a general comment, it is usually inappropriate to run post-hoc tests following omnibus tests that do not reveal any significant differences (unless you have an a priori hypothesis about a specific test and the result/direction of it you’re anticipating based on previous literature, e.g.). E.g., Lines 231-233, and Lines 241-242. Please remove these tests from the text throughout the results.
o When discussing results from Kruskal-Wallace tests, please change the language to something to the effect of “a significant difference between groups” or “no significant difference between groups”, as these tests can’t directly evaluate whether an individual treatment or interaction is present for 2-way designs. (E.g., lines 235-236 and lines 239-240, but please correct throughout). In many cases, the current language is confusing because “a significant effect of ASF and metyrapone treatments were detected” is followed a few sentences later by “post-hoc tests uncovered an effect of ASF, whereas metyrapone had no effect”.
o In general, I think it would make the text more readable to include interactions when discussing the models (such as “there was no effect of ASF, metyrapone, or their interaction on gene expression”, followed by all relevant statistics).
o Line 236-237: I think that this Tukey sentence is mistakenly placed here.
o The EWAT results starting on line 244 are hard to follow for the reader and should be reorganized.
o Line 272: Tukey test text is missing from this section
o Line 290: The results in this paragraph appear to contradict one another at times. Please correct.
o Line 309: The results in this paragraph appear to contradict one another, first saying the Mann Whitney test detected no differences, but then describing significant effects of the test on the final sentence of the paragraph.
• The tables and statistical results should be meticulously double-checked. For example, in the metyrapone table, in the spleen row, interaction column, it’s reported that there’s an interaction effect for IL-1beta. But this is not reported in the results section, and a KW test is used here, which wouldn’t provide an interaction.
• Line 355: I am not sure that quantifying the number of interactions across all tests accurately reflects the authors’ findings. For example, in some cases the interactions have more to do with the NSF-drug group, which would suggest the drug is having an effect only in the healthy sleepers – it’s unclear how to square that kind of finding with the original hypothesis the authors are interested in. Or, in other cases (like TNF-a in the liver in the mifepristone experiment), the control group is what’s elevated, indicating that drug treatment decreases expression and ASF doesn’t rescue the effect (i.e., a floor effect perhaps), which is again hard to interpret in the context of the authors’ original hypothesis.

Additional comments

This manuscript does a great job of tackling a complex set of results head on. The reporting is transparent, and the experimental design is solid. I also like their behavioral paradigm and that they’re careful about the sleep disturbance rate matching something people experience with severe sleep apnea. However, there are several places I had issues with understanding the methods or results, found text that appeared inconsistent with the data, and had some possible concerns with the statistics. I outline my criticisms in the other sections, and look forward to the authors’ responses!

Reviewer 2 ·

Basic reporting

This manuscript is well-written and the goals/hypotheses clearly articulated. The figures/tables are easily read, and the critical data are presented. All raw data are shared as supplemental material.

Relevant literature is not cited, specifically: studies by the group of Gozal demonstrating that chronic SF using this device increases TNFa activity; seminal works from the group of Krueger of the role pro-inflammatory cytokines play in the regulation of physiologic and pathologic sleep; and many studies demonstrating the role of the HPA axis in the regulation of wakefulness.

Experimental design

The experimental design is adequate and includes all appropriate controls. However,

- the rationale for acute vs chronic SF is not articulated. This is an important omission given the author's statements that insomnia and sleep apnea are sleep disorders that are characterized by SF. Both of these sleep disorders are chronic (acute insomnia, by definition, resolves and does not generally pose a public health risk).

- Literature is cited supporting the selection of the doses of GC inhibitor/antagonist used, but no data are presented as to circulating CORT concentrations at the 24h timepoint. The authors state in several places that short-half lives may be a contributing factor for the relatively modest changes in cytokine expression, and these statements could/should be supported by quantification of CORT from plasma samples.

- 0.9% saline was used as vehicle/control. Was this pyrogen-free saline?

Validity of the findings

- The conclusions are not overstated and reflect the findings presented; the statistical methods are sound and there are appropriate controls.

- Many studies of sleep-HPA axis-inflammatory mediators have been conducted, but few have antagonized GCs.

However,
- There are no statements concerning rigor and reproducibility.
- As stated above, there is no rationale for the use of acute rather than chronic SF within the context of insomnia and sleep apnea. Some studies demonstrate effects on inflammatory mediators when SF is chronic.

Additional comments

No additional comments.

---

## Round 0.2 · accepted · Accept

Dear Authors:

First of all, we greatly appreciate your patience. I have found his manuscript very interesting since the first time I read it, but some changes recommended by the reviewers were necessary. Thanks to these changes, the final manuscript has improved and become even more interesting. Therefore, I am pleased to inform you that your paper entitled: "Effect of glucocorticoid blockade on inflammatory responses to acute sleep fragmentation in Male mice" has been accepted for publication in PeerJ.

A cordial greeting and congratulations on your work.

Dr. Manuel Jimenez

·

Basic reporting

Typo line 424: regarding “the” hypothalamus

Table notes or caption: add what experimental group the arrows are increasing or decreasing in reference to (i.e., control condition).

Experimental design

No comment

Validity of the findings

No comment

Additional comments

Apart from my minor comment about the table caption/notes, the authors have adequately responded to my previous critiques and the manuscript is, in my opinion, okay for publication.

Reviewer 2 ·

Basic reporting

The authors have adequately addressed concerns/suggestions made by reviewers.

Experimental design

The authors provide a rationale for their inability to conduct additional assays/assessments. Some aspects of the study would be enhanced by additional experiments, but the manuscript contains enough novel data to warrant consideration for publication.

Validity of the findings

I have no additional comments.

Additional comments

I have no additional comments.